# Duration of Reduced CA19-9 Levels Is a Better Prognostic Factor Than Its Rate of Reduction for Unresectable Locally Advanced Pancreatic Cancer

**DOI:** 10.3390/cancers13164224

**Published:** 2021-08-22

**Authors:** Ko Tomishima, Shigeto Ishii, Toshio Fujisawa, Muneo Ikemura, Hiroto Ota, Daishi Kabemura, Mako Ushio, Taito Fukuma, Sho Takahashi, Wataru Yamagata, Yusuke Takasaki, Akinori Suzuki, Koichi Ito, Hiroaki Saito, Akihito Nagahara, Hiroyuki Isayama

**Affiliations:** Department of Gastroenterology, Graduate School of Medicine, Juntendo University, 3-1-3 Hongo, Bunkyo-ku, Tokyo 113-0033, Japan; tomishim@juntendo.ac.jp (K.T.); sishii@juntendo.ac.jp (S.I.); t-fujisawa@juntendo.ac.jp (T.F.); m-ikemura@juntendo.ac.jp (M.I.); h-ota@juntendo.ac.jp (H.O.); d.kabemura.nc@juntendo.ac.jp (D.K.); m-ushio@juntendo.ac.jp (M.U.); t.fukuma.vh@juntendo.ac.jp (T.F.); sho-takahashi@juntendo.ac.jp (S.T.); w.yamagata.mx@juntendo.ac.jp (W.Y.); ytakasa@juntendo.ac.jp (Y.T.); suzukia@juntendo.ac.jp (A.S.); kitoh@juntendo.ac.jp (K.I.); hiloaki@juntendo.ac.jp (H.S.); nagahara@juntendo.ac.jp (A.N.)

**Keywords:** CA 19-9, pancreatic cancer, prognostic factor

## Abstract

**Simple Summary:**

Upon diagnosis, about 35% of patients have initially unresectable locally advanced pancreatic cancer. The prognosis of these patients is still poor. Chemotherapy alone has been generally accepted as a standard therapeutic approach. However, clinical decision-making processes have not been established for aggressive treatments such as surgery and chemoradiotherapy in patients with a response and stable case of initially unresectable locally advanced pancreatic cancer. In the current study, we evaluated the reduction rate and duration of carbohydrate antigen (CA) 19-9 within 6 months as long-term survival. Cases of over 44% CA 19-9 reduction only one month from the baseline after treatment were not significantly associated with overall survival. On the other hand, more than 3 months of over 44% CA 19-9 reduction was significantly associated with prognosis, which is the same as the occurrence of distant metastasis. Multidisciplinary treatment focus on local treatment is expected in these selected patients.

**Abstract:**

A decrease in carbohydrate antigen (CA) 19-9 levels has been proposed as a prognostic marker for survival and recurrence in patients with pancreatic cancer. We evaluated the association between duration of reduced CA 19-9 levels during 6 months after treatment and long-term survival for 79 patients with unresectable locally advanced pancreatic cancer (LAPC). We calculated the differences between pretreatment and monthly CA19-9 levels. We categorized 71 patients with decreases in CA19-9 levels into three groups based on the duration of these reduced levels (>2, >3, and >4 months). The cut-off level for long-term (more than 2 years) survival was identified as a 44% reduction from the baseline, using a ROC curve. A reduction duration >2 months was not associated with overall survival (*p* = 0.1), while >3 months was significantly associated with survival (*p* =.04). In multivariate analysis, a reduction duration >3 months predicted a good long-term prognosis (odds ratio = 5.75; 95% confidence interval = 1.47–22.36; *p* < 0.01). In patients with unresectable LAPC, the duration of reduced CA19-9 levels for more than 3 months, rather than the rate of reduction in CA19-9 levels, during 6 months after treatment was significantly associated with good prognosis.

## 1. Introduction

Pancreatic cancer has an increasing incidence and the highest mortality out of all gastrointestinal cancers [1]. Complete surgical resection of pancreatic cancer is associated with the best outcomes. Unfortunately, less than 20% of patients are surgical candidates, and most have unresectable cancer [2]. Recent studies have demonstrated superior outcomes for surgery after neoadjuvant chemotherapy (NACT) compared to upfront surgery for patients with borderline resectable pancreatic cancer (BRPC) or locally advanced pancreatic cancer (LAPC) [3,4,5]. The optimal timing for surgery is 240 days after beginning NACT. Preoperative chemotherapy (CTx) or chemoradiotherapy (CRTx) prolongs survival in pancreatic cancer patients [6,7]. However, 20–70% of patients are surgical candidates after CTx or CRTx, and indications for surgery vary between institutions [8,9,10].

More than 80% of patients with advanced pancreatic cancer have increased levels of the serum tumor marker carbohydrate antigen (CA) 19-9 [11]. The levels of CA 19-9 reflect the dynamic effects of CTx or CRTx without patients negative for the Lewis blood group phenotype. CA 19-9 levels after NACT and surgical resection correlate with the R0 resection rate, histopathological response, and survival rates [12,13,14,15]. CA 19-9 levels also correlate with the survival of patients with advanced pancreatic cancer [16,17,18,19]. However, previous studies did not exclude patients with metastasis and used different CA 19-9 cut-off levels. Therefore, the optimal cut-off values for CA 19-9 levels are unclear. The cut-off values for CA 19-9 may be used to select LAPC patients with a good chance of survival for further treatment. In addition, prognostic factors for LAPC patients should be determined.

In this retrospective study, we determined the changes in CA 19-9 levels of unresectable LAPC patients during 6 months after CTx or CRTx.

## 2. Materials and Methods

### 2.1. Serum Levels of CA 19-9

CA 19-9 levels were measured monthly for 6 months after beginning treatment (upper limit: 37 U/mL). Patients negative for the Lewis blood group phenotype, defined as undetectable CA 19-9 throughout the study, were excluded [20]. In cases of obstructive jaundice, biliary drainage was performed prior to obtaining a serum sample. Changes in CA 19-9 levels from baseline were assessed using the reduction ratio, calculated as: current-CA 19-9 level/pretreatment CA 19-9 level (RR-C; reduction ratio of CA 19-9). For example, RR-C 3 months after the treatment was defined as: 3-month CA 19-9 level/pretreatment CA 19-9 level. Within normal limit in CA 19-9 levels after treatment was observed in 11 patients, who were included in the CA 19-9 decrease group.

### 2.2. Patients

We included 134 patients who were histologically diagnosed with unresectable LAPC, including initially BRCP, and received CTx or CRTx at Juntendo University Hospital between December 2005 and June 2020. We included initially BRPC cases that were progressed in spite of initial treatment. We excluded patients who received best supportive care (BSC) due to performance status of 3 or 4 (*n* = 5), being followed up for less than 3 months (*n* = 28), undergoing conversion surgery (*n* = 15), or being negative for the Lewis blood group phenotype (*n* = 7). After exclusion of these patients, 79 were included as finally unresectable locally advanced pancreatic cancer in the study (Figure 1). All cases of pathological phenotype were adenocarcinoma except one case of adenosquamous carcinoma. In addition, the relationships between CA 19-9 levels and long-term survival (defined as survival for >2 years) were also evaluated. Patients with a decrease in CA 19-9 levels (*n* = 71) were categorized into three groups based on the duration of CA19-9 reduction (i.e., 2, 3, or 4 months) (Figure 1). The medical records of patients were reviewed to obtain the age, sex, performance status, tumor location, CA19-9 levels (initial level, reduction rate, and reduction duration), tumor size, initial resectability, and treatment. The study protocol was approved by the Institutional Review Board of Juntendo University Hospital (IRB No: 20–007).

### 2.3. Resectability and Treatment

BRPC was defined as a tumor that was in contact with the common hepatic artery (CHA), without extension into the celiac axis (CA) or hepatic artery bifurcation; in contact with ≤180° of the circumference of the superior mesenteric artery (SMA); in contact with >180° of the circumference of the superior mesenteric vein or portal vein (PV); or in contact with ≤180° of the circumference of either the superior mesenteric or portal vein, with an irregular contour or venous thrombosis but possibility of reconstruction [21]. LAPC was defined as a tumor that had >180° contact with or invasion of SMA or PV and extended beyond the lower border of duodenum; >180° contact with or invasion of SMA or CA; or contact with or invasion of CHA, proper hepatic artery, CA, or aorta. These patients were not offered surgery because vascular invasion does not improve in the course of treatment.

The CTx or CRTx regimens were decided by the attending physician. The patients were evaluated using CA 19-9 levels and serial abdominal computed tomography (CT), magnetic resonance, and positron emission tomography (PET) imaging. CT images were evaluated using the modified Response Evaluation Criteria in Solid Tumors [22].

### 2.4. Statistical Analysis

The Kaplan–Meier method was used to estimate the overall survival (OS). OS was calculated based on the dates of diagnosis and death. Statistical differences in OS were analyzed using the two-tailed log-rank test. Fisher’s exact test was used to compare qualitative data, where appropriate. Logistic regression was used for multivariate analysis, after adjusting for all potential confounding factors. Differences were assumed to be significant at *p* < 0.05.

## 3. Results

### 3.1. Patient Characteristics

Table 1 summarizes the baseline patient characteristics. The median age of the study participants was 68 (36–87) years. There were 32 females (41%) and 47 males (59%). Most participants had a performance status of 0 (*n* = 63, 80%), while others had 1 or 2 (*n* = 16, 20%). The median pretreatment CA 19-9 was 182 (10–21,084) U/mL. There was no significant difference between initial CA 19-9 and overall survival (*p* = 0.45). The tumors were located in the pancreatic head (*n* = 41, 52%) or tail (*n* = 38, 48%), and the median tumor size was 32 (12–64) mm. The tumors were BRPC (*n* = 22, 28%) or LAPC (*n* = 57, 72%), with invasion into the CA (*n* = 33, 44%), SMA (*n* = 36, 45%), CHA (*n* = 3, 4%), or PV (*n* = 5, 7%). As for BRPC, 19 cases, in contact with ≤180° of artery, received chemotherapy, and three cases, in contact with >180° of portal vein, also received chemotherapy because of tumor diameter (more than 40 mm). CTx was administered to 63 patients (80%), and CRTx was given to 16 patients (20%). The CTx regimen consisted of gemcitabine plus nab-paclitaxel (*n* = 41), gemcitabine (*n* = 13), gemcitabine plus S-1 (*n* = 4, GS), S-1 (*n* = 3), or FOLFIRINOX (*n* = 2). CRTx patients received GS plus radiotherapy (GS-RT), with daily fractions of 1.8 Gy (total dose: 50.4 Gy over 5.5 weeks). Gemcitabine (200 mg/m^2^) was administered weekly for 6 weeks. S-1 (80 mg/m^2^/day) was administered orally twice a day on days 1–14 and 22–35. Progression-free survival and OS were 8 (3–74) and 17 (4–128) months, respectively.

### 3.2. CA19-9 Reduction Rate and Survival

CA19-9 decreased from the baseline for 71 (90%) patients on at least one occasion over the 6 months after treatment (RR-C < 0). Conversely, CA19-9 increased for eight (10%) patients (RR-C ≥ 1). RR-C < 1 was significantly associated with a better OS compared to RR-C ≥ 1 (median OS 21 m vs. 12 m, *p* = *0*.02, respectively). The CA19-9 cut-off value for long-term (>24 months) prognosis and RR-C were defined on the basis of 44% reduction from the ROC curve (75% of true positive fraction, 37% of false positive fraction, odds ratio 5.1). During 6 months after treatment, CA19-9 levels decreased for 71 patients, but improved OS was not observed in 64 (90%) of these patients who had RR-C >44% (median OS 21 m vs. 9 m, *p* = *0*.06) (Figure 2).

### 3.3. CA19-9 Reduction Duration and Overall Survival or Metastasis

Patients were categorized into three groups based on the duration of CA19-9 reduction (>2 months: *n* = 57, 80%; >3 months: *n* = 48, 68%; >4 months: *n* = 46, 65%) (Figure 1). We evaluated the relationships between these durations and OS. Reduction duration >2 months was not associated with OS, while >3 and >4 months were associated with OS (22 and 15 months, *p* = 0.04; 23 and 14 months, *p* = 0.01, respectively) (Figure 3). As for time to distant metastasis, we compared >3 months reduction and other cases. RR-C < 0.56 lasting for >3 months during the first 6 months after treatment predicted distant metastases (*p* = *0*.01) (Figure 4).

### 3.4. Factors Related to Long-Term Survival

Patients were divided into long-term (*n* = 22, LS group) and other survivors (*n* = 57, not LS group). Univariate analysis was performed for the effects of background factors and treatment methods on OS. Univariate analysis revealed that CRTx and CA 19-9 reduction >44% for >3 months was significantly associated with long-term OS (*p* = 0.03, *p* < 0.01) (Table 2). Multivariate analysis revealed that CA 19-9 reduction >44% for >3 months independently predicted OS (odds ratio 5.75, 95% CI 1.48–22.4, *p* = 0.01). CRTx was also associated with increased long-term OS, albeit without statistical significance (odds ratio 2.63, 95% CI 0.83–8.26, *p* = 0.09) (Table 2).

## 4. Discussion

Treatment of pancreatic cancer with distant metastases has improved with the use of chemotherapy. CA 19-9 is suitable for the dynamic assessment of effects of CTx or CRTx. Several studies have reported that CA 19-9 levels correlate with OS in patients with advanced pancreatic cancer. Hess et al. reported that pretreatment serum levels of CA 19-9 independently predict OS. However, in a large multicenter cohort study, a decrease in CA 19-9 levels 2 months after chemotherapy was not associated with OS [18]. The cohort study differed from our study in terms of chemotherapy regimen (gemcitabine and capecitabine in that study) and disease stage. Other studies have reported a positive association between improved OS with an early decrease in CA 19-9 levels in unresectable pancreatic cancer patients [16,17]. Treatment effects observed 8 weeks after the treatment predicted tumor control at 12 weeks and future treatment effects. The time interval from start of treatment to lowest CA 19-9 levels was longer in patients without metastases compared to those with metastases [19]. Previous studies included patients with advanced pancreatic cancer, with or without metastasis, but did not report long-term survival. CA 19-9 has recently been reported to be a predictor for better outcomes after NACT. In patients with LAPC who received NACT, pretreatment CA 19-9 levels did not affect OS, but post-treatment normalization of CA 19-9 predicted good outcomes [23,24]. Similarly, a sustained decrease in CA 19-9 levels predicted better outcomes in BRPC patients who received NACT [25].

The optimal treatment strategy for unresectable LAPC is controversial. We retrospectively studied 79 patients with unresectable LAPC, including initially BRCP, who were followed up for more than 3 months. BRPC is neither clearly resectable nor clearly unresectable but rather implies a greater chance of incomplete resection in the setting of upfront surgery. Additionally, many groups have proposed definitions; however, there is not yet a universally accepted definition of BRPC [26]. We included initially BRPC cases that were progressed in spite of initial treatment and categorized “unresectable cases after initial treatment” as “unresectable locally advanced pancreatic cancer” in this study. Progression-free survival and OS were 8 and 17 months, respectively, and similar to those in patients treated with gemcitabine plus S-1 therapy for LAPC (11.76 m vs. 16.41 m) [27]. CA 19-9 levels are affected by cholangitis or cholestasis. Therefore, we measured the CA 19-9 levels after treating these, after serum levels of bilirubin were within the normal range (<2.0 mg/dL).

Analysis of the RR-C changes in patients who survived <12, 12–24, or >24 months demonstrated that CA 19-9 levels decreased the most in the first 6 months (Figure 5). In patients who survived <12 months, CA 19-9 levels increased in the early stage, suggesting that the chemotherapy was ineffective. In patients who survived 12–24 months, CA 19-9 levels decreased 2 months after the treatment but increased again almost 4 months after the treatment. In the long-term survival group (>24 months), most patients had a decrease in CA 19-9 levels from baseline, and no patients had RR-C >1 at 6 months. Therefore, patients with a rapid increase in CA 19-9 levels following an initial decrease after the initiation of chemotherapy had a poor prognosis. Long-term control of CA 19-9 levels predicts long-term prognosis. Patients with CA19-9 reduction >44% for >3 months during the first 6 months had a positive association with long-term survival and distant metastases.

Although surgery is the only curative treatment for pancreatic cancer, invasion of the main vessels by the tumor precludes surgery. Furthermore, patients at low risk for distant metastases can receive local treatments. CRTx is associated with better long-term survival compared to CTx for LAPC patients, despite an increase in the treatment-related toxicities [28]. Distant metastases were reported after the first progression in 50–90% of patients who received CRTx [29,30]. Therefore, patients who have better long-term survival and low risk for distant metastases should be selected for CRTx. If the CA19-9 levels are reduced by >44% for >3 months, distant metastases are unlikely, and long-term survival (>24 months) can be expected. Multidisciplinary local treatments are expected to benefit these patients.

There were some limitations to our study. First, it was a single-center retrospective study, without a control group. Second, the sample size was small. Third, the CTx and CRTx regimens were not uniform. Finally, 43 follow-up CA 19-9 values (9%) were missing.

## 5. Conclusions

In conclusion, a duration of reduced CA19-9 levels >3 months, rather than the rate of reduction of CA19-9 levels, during the 6 months after treatment is significantly associated with a good prognosis and predicts long-term survival. We suggest that more aggressive treatment, such as surgery and radiotherapy, will be beneficial for these patients.

## Figures and Tables

**Figure 1 cancers-13-04224-f001:**
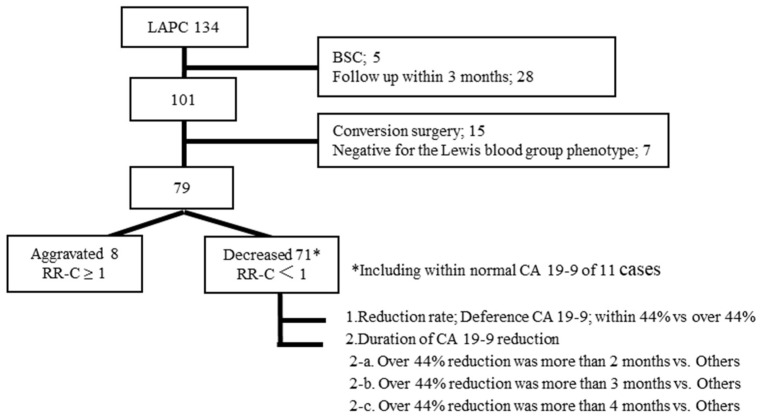
Patients flow chart. LAPC: locally advanced pancreatic cancer; BSC: best supportive care; RR-C: reduction ratio of CA 19-9.

**Figure 2 cancers-13-04224-f002:**
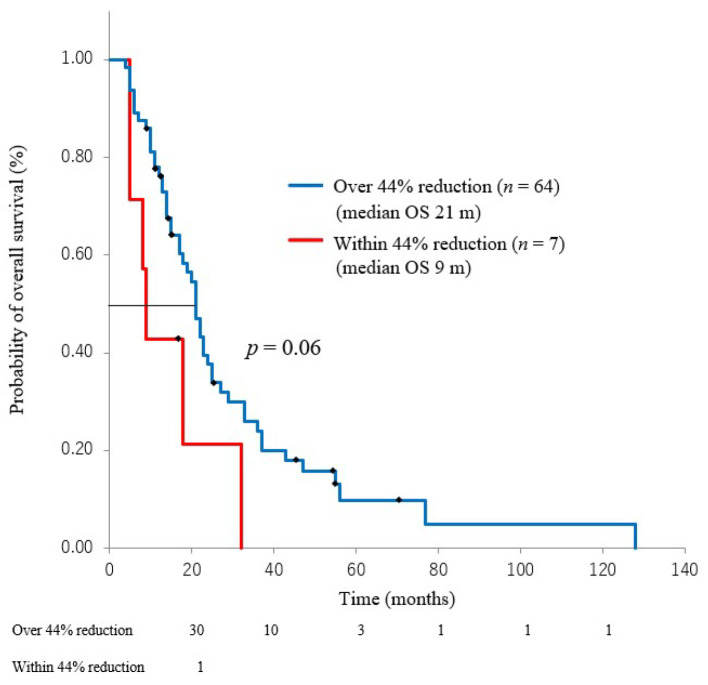
Among CA 19-9 reduction, relationship between within 44% reduction and over 44% reduction.

**Figure 3 cancers-13-04224-f003:**
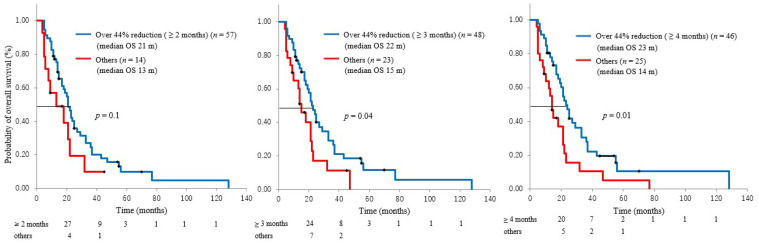
Among CA 19-9 reduction, relationship between >44% reduction of these durations (>2 months, >3 months, and >4 months) and OS.

**Figure 4 cancers-13-04224-f004:**
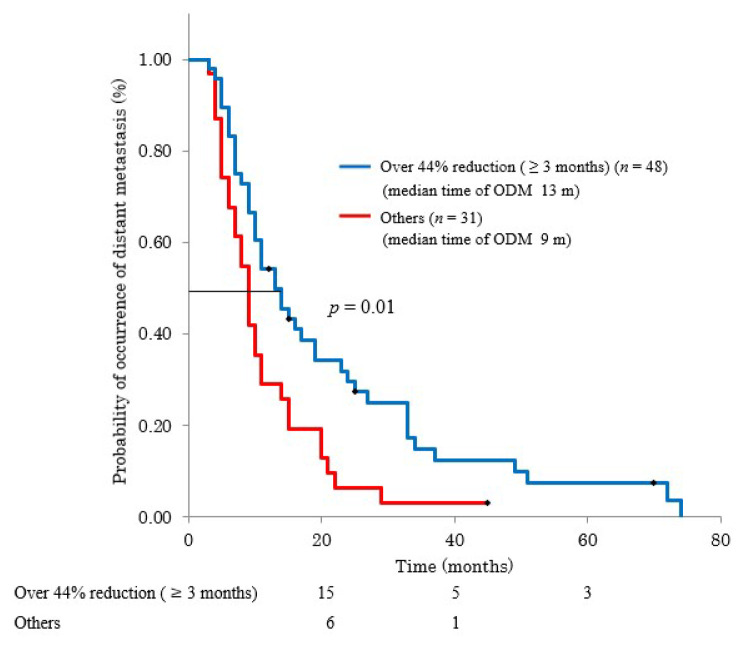
Relationship between >44% reduction duration (>3 months) and ODM. (ODM: occurrence of distant metastasis).

**Figure 5 cancers-13-04224-f005:**
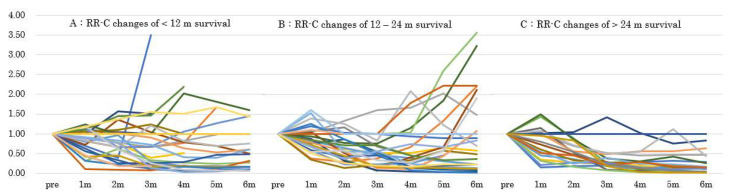
Analysis of the reduction ratio of CA19-9 (RR-C) in each survival. (Patients who survive within 12 months (**A**), from 12 months to 24 months (**B**), and more than 24 months (**C**).)

**Table 1 cancers-13-04224-t001:** Patient characteristics (*n* = 79).

Factors	*n* (%)	Median (Range)
Age, years		68 (36–87)
Sex, female/male	32 (41)/47 (59)	
PS, 0/1,2	63 (80)/16 (20)	
Pre-CA19-9, U/mL,		182 (10–21084)
Location, Head/Body-tail	41 (52)/38 (48)	
Tumor size, mm		32 (12–64)
Resectability, BR/LA	22 (28)/57 (72)	
Invasion, (CA/SMA/CHA/PV)	35 (44)/36 (45)/3 (4)/5 (7)	
1st Treatment, CTx/CRTx	63 (80)/16 (20)	
CTx regimen	GnP 41, Gem 13, GS 4, S1 3, FFX 2	
CRTx regimen	Gem + S1 + RT	
PFS, month		8 (3–74)
OS, month		17 (4–128)

PS: performance status; BR: borderline resectable, locally advanced; CA: celiac axis; SMA: superior mesenteric artery; CHA: common hepatic artery; PV: portal vein; CTx: chemotherapy; CRTx: chemoradiotherapy; PFS: progression free survival; OS: overall survival; GnP: gemcitabine plus nab-paclitaxel; Gem: gemcitabine; GS: gemcitabine plus S-1; FFX: FOLFIRINOX; RT: radiotherapy.

**Table 2 cancers-13-04224-t002:** Univariate and multivariate analysis for long term survival more than 2 years.

Variable		Univariate Analysis			Multivariate Analysis		
		LS (*n* = 22)	not LS (*n* = 57)	*p* Value	Odds Ratio	95% CI	*p* Value
Age (year)				0.80			
	≥65/<65	14/8	38/19				
Gender				0.30			
	Male/Female	15/7	32/25				
Performance status				0.07	0.35	0.07–1.81	0.21
	≥1/0	2/20	16/41				
Location				0.80			
	Head/Body-Tail	12/10	29/28				
Tumor Size (cm)				0.80			
	≥3/<3	14/8	35/22				
Initial Resectability				0.9			
	BR/LA	6/16	16/41				
Treatment				0.03	2.63	0.83–8.26	0.09
	CRTx/CTx	10/12	12/45				
CA19-9							
Initial level				1.00			
	≥37/<37	18/4	45/12				
Reduction cases				0.09			
	≥1/<1	0/22	8/49				
Duration of 44% reduction				<0.01	5.75	1.47–22.36	0.01
	≥3 m/Others	19/3	29/28				

## Data Availability

Please refer to suggested Data Availability Statements at https://www.mdpi.com/journal/cancers/instructions#suppmaterials (accessed on 23 July 2021).

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
