# Peer review of "Duration of Reduced CA19-9 Levels Is a Better Prognostic Factor Than Its Rate of Reduction for Unresectable Locally Advanced Pancreatic Cancer"

_cancers, 2021, doi:10.3390/cancers13164224_

Round 1
Reviewer 1 Report
The manuscript from Dr Tomishima et al investigated the correlation between duration of reduced CA 19-9 levels during 6 months after treatment and long-term survival of 79 patients with unresectable locally advanced pancreatic cancer. The authors found reduction duration > 3 months was significantly correlated with survival. In 23 multivariate analysis, reduction duration > 3 months predicted a good long-term prognosis. These findings suggest that duration of reduced CA19-9 levels for more than 3 months during 6 months after treatment may serve as an independent prognostic marker for unresectable locally advanced pancreatic cancer patients.
The manuscript provided significant new information. The experimental procedures are precisely described, the results are clearly presented, and the interpretation of the data is accurate. Although, there could be more sufficient background introduction and discussion of the studies on CA 19-9 at a molecular level in the context of pancreatic cancer. Also, it would greatly improve the novelty of this study if the authors could provide some mechanistic analysis.
A few minor comments include: 1. it is unclear what the initial CA19-9 levels range are before treatment and if the initial CA19-9 levels made a difference in the duration of CA19-9 reduction and patient survival. 2. What about the pathological diagnosis of the PDAC? Is there any correlation between prognosis and patient PADC pathological phenotype. Above clinical information would be greatly appreciated if being included in the paper.
Author Response
Cancers
Dear Assigned Editor Ms. Fannie Lin and Reviewer
Thank you very much for reviewing our manuscript and offering valuable advice. We have addressed your comments with point-by-point responses, and revised the manuscript accordingly. Please find the revised version of the manuscript entitled “cancers-1332010. Duration of Reduced CA19-9 Levels is a Better Prognostic Factor than Its Rate of Reduction for Unresectable Locally Advanced Pancreatic Cancer” with tables and figures to be considered for publication in Cancers.
Please contact me if there are further questions regarding this revised manuscript. We appreciate if decision of acceptance on this manuscript would be transferred by e-mail. Thank you for your consideration. We are looking forward to hearing from you.
Sincerely,
Ko Tomishima, M.D.
Hiroyuki Isayama, M.D., Ph.D.
Department of Gastroenterology,
Juntendo University, School of Medicine,
2-1-1, Hongo, Bunkyo-ku, Tokyo,
113-8421, Japan
Phone; +81-3-5802-1060
Fax; +81-3-3813-8862
E-mail; tomishim@juntendo.ac.jp
Manuscript IDï¼›cancers-1332010
Reviewer 1
The manuscript from Dr Tomishima et al investigated the correlation between duration of reduced CA 19-9 levels during 6 months after treatment and long-term survival of 79 patients with unresectable locally advanced pancreatic cancer. The authors found reduction duration > 3 months was significantly correlated with survival. In 23 multivariate analysis, reduction duration > 3 months predicted a good long-term prognosis. These findings suggest that duration of reduced CA19-9 levels for more than 3 months during 6 months after treatment may serve as an independent prognostic marker for unresectable locally advanced pancreatic cancer patients.
The manuscript provided significant new information. The experimental procedures are precisely described, the results are clearly presented, and the interpretation of the data is accurate. Although, there could be more sufficient background introduction and discussion of the studies on CA 19-9 at a molecular level in the context of pancreatic cancer. Also, it would greatly improve the novelty of this study if the authors could provide some mechanistic analysis.
A few minor comments include:
- it is unclear what the initial CA19-9 levels range are before treatment and if the initial CA19-9 levels made a difference in the duration of CA19-9 reduction and patient survival.
→Thank you for valuable advice. We corrected it as follow; “The median pretreatment CA 19-9 was 182 (10–21,084) U/mL. There was no significant difference between initial CA 19-9 and overall survival (p=.45).” (P3. Lines 142-143). We considered there is no correlation between initial CA 19-9 and OS for progressive PDAC, unlike surgery cases. Treatment response (changes of CA 19-9 values) is more important than initial CA 19-9 values. Additionally, there is no relationship between initial CA 19-9 and duration of CA 19-9 reduction (≧ 3 months) (p=0.46). Hence, in this current study, as for UR-LA PDAC, initial CA 19-9 is no differences in the duration of CA 19-9 reduction and patient survival.
- What about the pathological diagnosis of the PDAC? Is there any correlation between prognosis and patient PADC pathological phenotype. Above clinical information would be greatly appreciated if being included in the paper.
→Thank you for your great suggestion. We corrected it as follow; “All cases of pathological phenotype were adenocarcinoma except one case of adenosquamous carcinoma.” (P2, Line 83-84). So, we don’t do statistical exam about pathological phenotype.

Reviewer 2 Report
I have with great interest read this original article regarding the prognostic values of a CA19-9 reduction for unresectable pancreatic cancer. This is an interesting article that have the potential to improve treatment of pancreatic cancer however, several questions arose when I read the manuscript.
- My first and more or less most important question arise already in the title “unresectable locally advanced pancreatic cancer”. Since the study clearly also involves borderline resectable pancreatic cancers and not only LAPC. It is of utmost importance to define whether a pancreatic cancer is BRPC or LAPC, i.e. do we have a curative potential or not. Is the study on LAPC or not?
- Q1 of course give rise to more questions, why did you exclude the patients that went to surgery? Since the goal of neoadjuvant treatment, especially in a fit cohort as your with 80 % performance status=0 and median age of 68, is curation, why not keep all patients, including the resected ones. The alternative, which I would not recommend, would be to exclude the BRPC from the beginning. If this cohort of “failed” BRPC and LAPC should be kept a thorough explanation is mandatory in the text.
- The definition of BRPC and LAPC is a bit outdated but I presume that this was the standard definition at the time of the start of inclusion. I would have liked to know how old the data is although the range of OS 4-128 gives some clues.
- You have treated jaundice before obtaining baseline CA199 which seems adequate but what did you do if jaundice arose during the CTx or CTRx did you then treat the jaundice before the new CA 19-9 value?
- I can’t find in the manuscript how often CA19-9 was checked, and how far was the median time from the intended date. This is important since we are drawing the conclusions based on the timing of the tests.
- Since English is not my native language I have not done a review of the language, but I find it easy to follow and a language check seems to have been done according to the added file of nonpublished material.
A revision of the paper is mandatory before publication.
Regards
Author Response
Aug 13, 2021
Cancers
Dear Assigned Editor Ms. Fannie Lin and Reviewer
Thank you very much for reviewing our manuscript and offering valuable advice. We have addressed your comments with point-by-point responses, and revised the manuscript accordingly. Please find the revised version of the manuscript entitled “cancers-1332010. Duration of Reduced CA19-9 Levels is a Better Prognostic Factor than Its Rate of Reduction for Unresectable Locally Advanced Pancreatic Cancer” with tables and figures to be considered for publication in Cancers.
Please contact me if there are further questions regarding this revised manuscript. We appreciate if decision of acceptance on this manuscript would be transferred by e-mail. Thank you for your consideration. We are looking forward to hearing from you.
Sincerely,
Ko Tomishima, M.D.
Hiroyuki Isayama, M.D., Ph.D.
Department of Gastroenterology,
Juntendo University, School of Medicine,
2-1-1, Hongo, Bunkyo-ku, Tokyo,
113-8421, Japan
Phone; +81-3-5802-1060
Fax; +81-3-3813-8862
E-mail; tomishim@juntendo.ac.jp
Manuscript IDï¼›cancers-1332010
Reviewer
I have with great interest read this original article regarding the prognostic values of a CA19-9 reduction for unresectable pancreatic cancer. This is an interesting article that have the potential to improve treatment of pancreatic cancer however, several questions arose when I read the manuscript.
1 My first and more or less most important question arise already in the title “unresectable locally advanced pancreatic cancer”. Since the study clearly also involves borderline resectable pancreatic cancers and not only LAPC. It is of utmost importance to define whether a pancreatic cancer is BRPC or LAPC, i.e. do we have a curative potential or not. Is the study on LAPC or not?
→Thank you for valuable advice. We corrected it as follow; “The tumors were BRPC (n = 22, 28%) or LAPC (n = 57, 72%), with invasion into the CA (n = 33, 44%), SMA (n = 36, 45%), CHA (n = 3, 4%), or PV (n = 5, 7%). As for BRPC, 19 cases, in contact with ≤ 180° of artery, received chemotherapy, and 3 cases, in contact with > 180° of portal vein, also received chemotherapy because of tumor diameter (more than 40mm).” (P4, Line 145-149). BRPC was categorized as unresectable before. Re-evaluating old cases, 22 cases of unresectable PDAC resulted in BRPC.
2 Q1 of course give rise to more questions, why did you exclude the patients that went to surgery? Since the goal of neoadjuvant treatment, especially in a fit cohort as your with 80 % performance status=0 and median age of 68, is curation, why not keep all patients, including the resected ones. The alternative, which I would not recommend, would be to exclude the BRPC from the beginning. If this cohort of “failed” BRPC and LAPC should be kept a thorough explanation is mandatory in the text.
→This is also valuable advice. Thank you very much for your suggestion. Our data span was so long (from December 2005 to June 2020). So BRPC was treated as unresectable before. We categorized “finally unresectable cases” as “unresectable locally advanced pancreatic cancer”. We corrected it as follow; “We included 134 patients who were histologically diagnosed with unresectable LAPC, including BRCP, and received CTx or CRTx in Juntendo University Hospital between December 2005 and June 2020.” (P2, Line 77-79). And “After exclusion of these patients, 79 were included as finally unresectable locally advanced pancreatic cancer in the study.” (P2. Line 82-83).
3 The definition of BRPC and LAPC is a bit outdated but I presume that this was the standard definition at the time of the start of inclusion. I would have liked to know how old the data is although the range of OS 4-128 gives some clues.
→This data was in Juntendo University Hospital between December 2005 and June 2020. (P2, Line 77-79).
4 You have treated jaundice before obtaining baseline CA199 which seems adequate but what did you do if jaundice arose during the CTx or CTRx did you then treat the jaundice before the new CA 19-9 value?
→Thank you for reviewers suggestion. We chose CA 19-9 values at the time of well-treated jaundice. If impossible, CA 19-9 values at the time were handled as blank. (P8, Line 251).
5 I can’t find in the manuscript how often CA19-9 was checked, and how far was the median time from the intended date. This is important since we are drawing the conclusions based on the timing of the tests.
→Thank you for reviewers suggestion. We suggested it “CA 19-9 levels were measured monthly for 6 months after beginning treatment (upper limit: 37 U/mL).” (P2, Line 67-68).
6 Since English is not my native language I have not done a review of the language, but I find it easy to follow and a language check seems to have been done according to the added file of nonpublished material.
→Yes. This manuscript was checked and corrected the English by Textcheck. (certified@textcheck.com) (Please quote reference number: '21071217')

Round 2
Reviewer 2 Report
I have with great interest read the revised manuscript. I think it has improved since the last time I read it. Most of my questions have been answered. Regarding the question below the authors have answered the question to me but I think that the manuscript would improve further if the text also clearly highlight the reason for including booth LAPC and BRPC in the cohort.
Besides this I recommend publication
Regards
2 Q1 of course give rise to more questions, why did you exclude the patients that went to surgery? Since the goal of neoadjuvant treatment, especially in a fit cohort as your with 80 % performance status=0 and median age of 68, is curation, why not keep all patients, including the resected ones. The alternative, which I would not recommend, would be to exclude the BRPC from the beginning. If this cohort of “failed” BRPC and LAPC should be kept a thorough explanation is mandatory in the text.
→This is also valuable advice. Thank you very much for your suggestion. Our data span was so long (from December 2005 to June 2020). So BRPC was treated as unresectable before. We categorized “finally unresectable cases” as “unresectable locally advanced pancreatic cancer”. We corrected it as follow; “We included 134 patients who were histologically diagnosed with unresectable LAPC, including BRCP, and received CTx or CRTx in Juntendo University Hospital between December 2005 and June 2020.” (P2, Line 77-79). And “After exclusion of these patients, 79 were included as finally unresectable locally advanced pancreatic cancer in the study.” (P2. Line 82-83).
Author Response
Aug 16, 2021
Cancers
Dear Assigned Editor Ms. Fannie Lin and Reviewer
Manuscript IDï¼›cancers-1332010 (Reviewer 2)
I have with great interest read the revised manuscript. I think it has improved since the last time I read it. Most of my questions have been answered. Regarding the question below the authors have answered the question to me but I think that the manuscript would improve further if the text also clearly highlight the reason for including both LAPC and BRPC in the cohort.
→Thank you very much for your additional advice.
We corrected it as “We included initially BRPC cases who were progressed in spite of initial treatment.” (P2, Line 79-80). And “We retrospectively studied 79 patients with unresectable LAPC including initially BRCP, who were followed up for more than 3 months. BRPC is neither clearly resectable nor clearly unresectable but rather implies a greater chance of incomplete resection in the setting of upfront surgery. And many groups have proposed definitions, however there is not yet a universally accepted definition of BRPC [26]. We included initially BRPC cases who were progressed in spite of initial treatment, and categorized “unresectable cases after initial treatment” as “unresectable locally advanced pancreatic cancer” in this study.” (P7 Line218-225).
Sincerely,
Ko Tomishima, M.D.
Hiroyuki Isayama, M.D., Ph.D.
Department of Gastroenterology,
Juntendo University, School of Medicine,
2-1-1, Hongo, Bunkyo-ku, Tokyo,
113-8421, Japan
Phone; +81-3-5802-1060
Fax; +81-3-3813-8862
E-mail; tomishim@juntendo.ac.jp
